# Current Advances in 3D Bioprinting for Cancer Modeling and Personalized Medicine

**DOI:** 10.3390/ijms23073432

**Published:** 2022-03-22

**Authors:** Nicolas Germain, Melanie Dhayer, Salim Dekiouk, Philippe Marchetti

**Affiliations:** 1UMR 9020–UMR-S 1277–Canther–Cancer Heterogeneity, Plasticity and Resistance to Therapies, Institut de Recherche Contre le Cancer de Lille, University Lille, CNRS, Inserm, CHU Lille, F-59000 Lille, France; melanie.dhayer@inserm.fr (M.D.); salim.dekiouk@inserm.fr (S.D.); 2Banque de Tissus, Centre de Biologie-Pathologie, CHU Lille, F-59000 Lille, France

**Keywords:** 3D bioprinting, 3D printing, bioink, cancer, cell biology

## Abstract

Tumor cells evolve in a complex and heterogeneous environment composed of different cell types and an extracellular matrix. Current 2D culture methods are very limited in their ability to mimic the cancer cell environment. In recent years, various 3D models of cancer cells have been developed, notably in the form of spheroids/organoids, using scaffold or cancer-on-chip devices. However, these models have the disadvantage of not being able to precisely control the organization of multiple cell types in complex architecture and are sometimes not very reproducible in their production, and this is especially true for spheroids. Three-dimensional bioprinting can produce complex, multi-cellular, and reproducible constructs in which the matrix composition and rigidity can be adapted locally or globally to the tumor model studied. For these reasons, 3D bioprinting seems to be the technique of choice to mimic the tumor microenvironment in vivo as closely as possible. In this review, we discuss different 3D-bioprinting technologies, including bioinks and crosslinkers that can be used for in vitro cancer models and the techniques used to study cells grown in hydrogels; finally, we provide some applications of bioprinted cancer models.

## 1. 3D Bioprinting at a Glance

### 1.1. Introduction

Additive manufacturing has been a major breakthrough in construction technologies and has been considered “the third industrial revolution” [1]. Additive manufacturing, commonly known as 3D printing, allows for building parts one layer at a time from a 3D computer model, allowing for rapid design optimization and customization. Because of these interesting properties, medical applications have been quickly developed: 3D-printed prostheses, implants, anatomical models, etc. [2,3]. The ease of use and speed of prototyping has even allowed for quick responses to medical needs during the COVID-19 pandemic [4,5,6].

The rapid development of this technology has required the development of new materials capable of being printed, in particular plastics, but also metals, ceramics, and elastomers. Traditionally, the materials used for 3D printing in medicine are made of inert and acellular materials, such as plastics [7]. Among those materials, some are bio-compatible and can thus be used for implantation [8]; other materials are degradable and are used as guides for soft tissue reconstruction, e.g., breast reconstruction after cancer surgery [9]. Recently, a new field of research in 3D printing has emerged: 3D bioprinting. Three-dimensional bioprinting uses 3D-printing technology to print cells and a supportive matrix (called bioink) altogether, ultimately printing a living tissue [10]. Bioinks have been defined by Groll et al. as “a formulation of cells suitable for processing by an automated biofabrication technology that may also contain biologically active components and biomaterials” that could be resumed as cell-containing materials [11]. Three-dimensional bioprinting, while promising, raises a large number of concerns and challenges, in particular the development of biocompatible bioinks and their integration into the human body; however, it seems to be a proper tool for complex tissue in in vitro modeling [12]. Although bioprinting is mainly used for tissue engineering, it seems to us that this technology also has its place in research teams wishing to model the complexity of the cancer microenvironment: its heterogeneity, its mechanical environment, its metabolism, and its neoangiogenesis, all of which can, of course, be used for drug-screening purposes.

In this review, we will try to propose an easy approach that allows the implementation of bioprinting in a research team and more particularly in the context of cancer modeling. We will first summarize the main technical features necessary for the implementation of bioprinting: the choice of the bioprinter, the choice of the bioink, and the polymerization method. Secondly, we will detail how bioprinting allows us to refine cancer research, notably by adding a cellular and mechanical complexity that 2D culture cannot provide. The aim is not to be exhaustive but rather to be comprehensive.

### 1.2. Bioprinting Technologies

The first bioprinting technique was described in 2003 by Boland et al., who used an inkjet-based technique to print 2D tissue constructs [13]. Since this first experiment, numerous bioprinting technologies have been created and can be classified into three main categories depending on the type of cell deposition: drop-based (e.g., inkjet or laser bioprinting), filament-based (e.g., extrusion bioprinting), and plane-based (e.g., digital light processing (DLP)/stereolithography (SLA) bioprinting) (Table 1).

Nowadays, the most-used technology is the filament-based one, with different extrusion mechanisms: pneumatic, piston, and screw-driving (Figure 1). Extrusion-based techniques resulting in filament deposition are nowadays the most used as they can quickly produce scaffolds of a resolution down to 100 μm in an affordable and relatively simple way [14].

In our opinion, nowadays, this technology is the easiest to implement; many manufacturers offer machines with multiple extrusion printheads (some printheads may even use inkjet-based printing techniques (see below for details)) all in a tabletop format and with a user-friendly interface at reasonable prices. This technology is also compatible with almost all bioink formulations [15].

Droplet-based techniques are consistent with the discontinuous printing of microdroplets and thus a high resolution (for review, see [16]). Inkjet printing is the most common technology used for droplet generation and consists of a piezoelectric or thermal actuator that allows the precise deposition of the droplets down to 50 μm [17,18]. Laser-based droplet deposition allows single-cell deposition and, as a non-contact method, is responsible for low shear stress and thus excellent viability; the drawback is the expensive price of this type of 3D printer [19,20]. There are also other less-used approaches, such as acoustic- or valve-based droplet bioprinting technologies [21,22]. Even if the droplet generation (surface tension) and breaking combined with the force with which it will be projected onto the printing plate can reduce cell viability, drop-based approaches allow higher cell viability than filament-based ones (>85%).

This technology, although it brings a precision that extrusion-based ones cannot have, only allows the printing of 2D patterns. This can be useful to precisely include cells in a pre-existing 3D matrix but not for large-scale constructions.

Plane-based 3D printing is mainly consistent in DLP and SLA technology (for review, see [23,24]). In SLA technology, photopolymerization is achieved through a laser beam scanning the surface of a liquid bioink, whereas in the DLP technique, polymerization is achieved by a digital micromirror device (DMD) or by a liquid crystal display (LCD) [25]. Volumetric bioprinting is a technique derived from those light-based techniques and can enable the creation of entire objects at once, which allows free-form architecture bioprinting that cannot be achieved with other technologies [26].

Those techniques have a high resolution down to 25 μm and speed in producing large and complex volumes; however, this technique requires a large volume of bioink, a significant part of which will not be polymerized [27].

Despite these interesting characteristics, particularly the speed of printing large volumes and the precision, it is not the easiest to implement this technology, particularly because of the lack of compatible bioinks and its running cost. It is, however, interesting for printing complex microfluidic structures.

### 1.3. Bioinks

As described above, bioinks are a blend of biomaterials (mainly hydrogels) and cells (Table 2). Properties of such biomaterials are thus critical to their printability and the cells’ biological requirements. Biomaterials can be separated into two categories: those derived from naturally occurring products and those of synthetic origin [28]. Their mechanical properties and biodegradation should be carefully assessed to assure (i) printability that will assure easy printing and high shape fidelity, (ii) that cell death will not be induced during the printing phase, (iii) a construct with the desired mechanical strength, and (iv) long-term biocompatibility once printed [29,30]. Using a biomaterial with minimal batch-to-batch variation is also essential to have reproducible 3D constructions. Hydrogels, which are mainly composed of water, are thus the most adapted bioinks, as they show good capability for mimicking the natural cell environment combined with good biodegradability and printability [31]. As reported above, the two categories, natural and synthetic, show different properties; natural hydrogels show limited mechanical strength and are subject to rapid degradation, whereas synthetic biomaterials show poor biocompatibility but good mechanical properties and printability. Therefore, some authors mix the two to obtain a “hybrid bioink.” 

The choice of bioink is therefore based on (i) the choice of bioprinting technology, (ii) the characteristics (stiffness, microenvironment) of the tissue to mimic, (iii) the need for a shape fidelity, and (iv) the crosslinking technology available. To facilitate the choice of the bioink, a table summarizing the main bioinks and their characteristics is provided (Table 2).

### 1.4. Crosslinking

One of the key steps in the bioprinting process is the crosslinking procedure, as it plays a crucial role in the stabilization of the 3D construct and thus in cell viability (for review, see [32]). Crosslinking results in chemical and physical modification of the bioinks that will ensure the stability of the different printed layers altogether. Depending on the bioink’s nature, crosslinking can be achieved through enzymatic (e.g., fibrinogen thrombin with fibrin-based bioinks), ionic (e.g., CaCl2 with alginate), chemical (e.g., horseradish peroxidase and alginate), physical (e.g., UV with gelatin methacrylate), and thermal (e.g., gelatin) processes. Crosslinking can be achieved before printing, during the printing process, or after printing (Figure 1). 

As with bioinks, the choice of crosslinking method will mainly be based on (i) the different crosslinking methods available for the bioink chosen, (ii) the necessity of high cell viability after printing (UV light, for example, is widely spread as a crosslinker but can induce DNA damage resulting directly from light radiation but which may also be related to toxicity linked to the photoinitiator), and (iii) the need for constant mechanical properties over time.

As briefly reported above, there are numerous bioprinting technologies, many biomaterials that can be turned into bioinks, and many ways to crosslink, which results in a large number of possibilities. Those many possibilities are a strength on the one hand, as it is, therefore, a tunable technology, but, on the other hand, they require a detailed protocol to obtain a satisfactory result. Moreover, despite many publications concerning bioprinting, few protocols are sufficiently detailed to be properly reproduced. We therefore propose a process flow applicable to all filament-based bioprinting with the main parameters to be taken into account and the cell parameters to be assessed at each stage (Figure 2). Bioprinting is, however, a major breakthrough in tissue engineering and in vitro complex modeling and appears to be a tool of choice for the study of cancer, as will be described in the following paragraphs. We will first describe how to analyze bioprinted constructs and then detail why this technology seems particularly adapted to the study of cancer.

**Table 1 ijms-23-03432-t001:** Most commonly used bioprinting technologies.

Type of Technology	Example of Printing Method	Advantages	Disadvantages	Cell Density	Average Cell Viability	Crosslinking	References
Droplet-based	Laser	Very high accuracy and resolution Low shear stress Very expensive	Only low-viscosity bioinks Only 2D patterns (limited high)	Low (less than 10 million per mL)	High	Depends on biomaterial used	[19,20]
	Inkjet	High accuracy Low shear stress					[17,18]
Filament-based	Worm drive Pneumatic Syringe/piston	Large panel of bioinks available Low cost Highly tunable	Higher shear stress and lower cell viability than other bioprinting technologies	High (more than 10 million per mL)	Medium/high depending on nozzle and pressure	Depends on biomaterial used	[13,14,15,33]
Plane-based/Volumetric	DLP/SLA	Fast for large and complex 3D models Very high accuracy	Few bioinks available Waste of bioink due to its conception	High (more than 10 million per mL)	High	Photocurable by DLP/SLA technology	[23,24,25,26,27]

**Table 2 ijms-23-03432-t002:** Examples of bioinks and their applications in cancer research.

	Material	Type of Bioink	Bioprinting Technology	Tissue Engineering Model	Cancer Models	Advantages	Drawbacks	Type of Crosslinking	References
Bioink derived from natural biomaterials	Alginate-based	Natural polysaccharide (brown algae)	Drop-based Filament-based	Vascular, cartilage, bone, neural tissue, fibroblast, and many more	Drug delivery Cancer stem cell research Breast cancer, melanoma, and many more cancers Tumor spheroids	Low cost Good printability Excellent bio-compatibility	Poor cell adhesion Fast degradation	Ionic	[34,35,36,37,38,39,40]
	Gelatin-based	Natural protein (bovine skin and tendon)	Drop-based Filament-based Plane-based	Vascular, cartilage, bone, muscle, fibroblast, and many more	Cholangiocarcinoma, bladder cancer, and many more cancers Tumor spheroids	Excellent bio-compatibility Low-cost High cellular adhesion	Low viscosity at room or higher temperatures Need a temperature-controlled (cooled printhead) and a cooled printbed Low mechanical strength (higher if blended with methacrylate)	Chemical Thermal UV Covalent Enzymatic	[38,41,42,43,44,45,46]
	Cellulose and nanocellulose-based	Natural polysaccharide obtained from the biosynthesis of plants or bacteria	Filament-based	Cartilage and bone	Drug delivery Gastric, cervical, pancreatic, and many more cancers	Great similarity with ECM Excellent bio-compatibility	Low viscosity for cellulose nanocrystals Mainly used mixed with other natural biomaterials	Enzymatic UV	[47,48,49,50,51]
	Matrigel	Solubilized basement membrane matrix secreted by Engelbreth-Holm-Swarm (EHS) mouse sarcoma cells	Filament-based Drop-based	Vascular, liver, bone, lung, and many more	Tumor spheroids Many types of cancer	Most used material in cancer research Excellent bio-compatibility Very well characterized for organoid/spheroid formation	Cannot be used alone due to its complex rheological behavior and low mechanical properties Limited use in vivo due to its mouse tumor origin Expensive High batch variability	Thermal	[52,53,54,55,56]
	Collagen-I-based	Natural protein (rat tail or bovine skin and tendon)	Drop-based Filament-based	Hard tissues (bone, osteochondral, cartilage) Skin, cardiovascular, and liver tissues; nervous system; and cornea	Tumor spheroids Neuroblastoma, breast cancer	Excellent bio-compatibility High cellular adhesion Minimal immunogenicity Excellent printability Enzymatically degradableMechanical and structural properties close to native tissue	Low shape fidelity	pH Thermal	[57,58,59,60]
	Hyaluronic-acid-based	Natural polysaccharide (bacterial fermentation or animal products)	Filament-based	Hard tissues (bone, osteochondral, cartilage)	Tumor spheroids Melanoma, breast cancer	Excellent bio-compatibility Highly tunable (wide variety and high degree of potential chemical modifications) Interact with cell receptors	Poor mechanical strength Mainly used mixed with other natural biomaterials	Depends on the other biomaterial/chemical modifications Physical or covalent	[61,62,63,64]
	Agarose-based	Natural polysaccharide derived from red seaweed	Filament-based	Bone, vascular, neural, and adipose tissue	Leukemia	Good biocompatibility Great similarity with ECM Thermo-reversible gelling	Poor cell survival if not blended with another biomaterial Poor printability (needs high temperature for dispensing (70 °C) and gels at low temperatures)	Thermal Ionic	[53,65,66]
	Fibrin-based	Natural protein (human plasma)	Filament-based Drop-based	Muscular, neural, skin, and adipose tissue, wound healing model	Drug release Glioblastoma	High shape fidelity (depending on fibrinogen–thrombin concentration) Excellent biocompatibility Enzymatically degradable	Medium cell adhesion Low mechanical properties	Enzymatic (fibrinogen–thrombin)	[67,68,69]
	Silk-derived	Natural protein (bombyx mory)	Filament-based	Hard tissues (bone, osteochondral, cartilage), vascular tissue	Drug delivery	High shape fidelity Low Cost Good biocompatibility	Lacks cell-binding domains Medium cell viability Needs other supportive material for cell proliferation (alginate, gelatin, etc.) Poor printability performance	Enzymatic Physical	[70,71,72,73]
	Gellan gum	Natural polysaccharide	Filament-based	Hard tissues (bone, osteochondral, cartilage), brain-like structures	Drug delivery	Excellent biocompatibility Low cost Rapid gelation	Poor printability performance	Thermal	[74,75,76]
	Chitosan	Natural polysaccharide produced by deacetylation of chitin (extract from shrimps)	Filament-based Drop-based Plane-based	Hard tissues (bone, osteochondral, cartilage), vascular, skin, and hepatic tissues	Drug delivery	Good biocompatibility Medium to high cell viability	Medium cell adhesion Low shape fidelity Low mechanical properties	Ionic UV	[77,78,79]
	Polypeptides	Corning (PuraMatrix)	Filament-based Droplet-based Plane-based	Liver, neural	Ovarian cancer	Self-assembly Adapted for soft-tissue applications and in conjunction with other materials	Low pH leading to low cell viability	Ionic-complementary self-assembly	[80,81]
	De-cellularized matrix-based (dECM)	Natural matrix	Filament-based	Adipose, hepatic, and heart tissues; MSCs; cancer models	Many tumor models depending on dECM	Renders natural ECM Tissue-specific	Low mechanical properties Protein denaturation during fabrication processes Poor printability if not mixed with another biomaterial Long procedure	Depends on the other biomaterial/chemical modifications	[82,83,84,85]
Bioink derived from synthetic biomaterials	AM (acrylamide)	Polyacrylamide	Filament-based Plane-based Droplet-based	Different stiffness models	Melanoma, breast cancer	Wide range of elasticity Most standardized protocol	Suitable for 2D culture only or necessary to couple it with another material	UV	[86,87]
	PCL/PLGA	Poly(caprolactone)/Poly(lactic–glycolic acid)	Filament-based Drop-based	Hard tissues (bone, osteochondral, cartilage)	Mainly depends on the natural biomaterial used	Good mechanical strength Controllable rate of degradation	Mainly used as a scaffold (melting temperature around 60 °C not compatible with cell viability) Needs other supportive material for cell proliferation (alginate, gelatin, etc.)	Depends on the natural biomaterial used	[88,89,90]
	PEG	Polymer of ethylene oxide	Filament-based	Vascular and bone tissue		Highly tunable (mechanical properties, polymerization, chemical composition)	Needs chemical modification to be printed Requires the addition of bioactive molecules to allow cellular interaction (high hydrophobicity)	UV if mixed with a photoinitiator Condensation Michael-type addition Click chemistry Native chemical ligation Enzymatic reaction	[91,92,93]
	Pluronic	Triblock copolymer of poly(ethylene glycol)-poly(propylene oxide)-poly(propylene glycol)	Filament-based	Cartilage		High shape fidelity Good printability	Lacks cell-binding domains Low cell viability Poor mechanical strength	Covalent	[94,95]
	PU	Polyurethane	Filament-based	Cartilage Neural stem cells		Good biocompatibility and biodegradability High mechanical strength	Needs other supportive material for cell proliferation (alginate, gelatin, etc.)	Depends on the natural biomaterial used	[96,97]

## 2. Characterization of Cells after Bioprinting

To evaluate the success of a bioprinting model, one of the most important parameters to assess is the viability and metabolic activity of the cells. Indeed, it is necessary to find the adequate printing parameters that allow for obtaining the structural integrity of the hydrogel so that it is reproducible and especially viable. These parameters must be determined for each type of bioink and even for each concentration. Printing parameters, such as the bed or cartridge temperature, pressure, and printing speed, will modify the viscosity of the gel, which will affect the shear stress exerted on the cells and, therefore, their viability. This is also impacted by the way the hydrogels are crosslinked. 

A plethora of techniques is available to characterize cells after bioprinting to determine the size and organization of the constructs, cell viability, and metabolism and the level of gene and protein expression [98,99] (Table 3, Figure 3). The size and shape of the constructs must be adapted to the technique used. For example, microscopic analysis does not require many cells, in contrast to cytometry, molecular biology technics, or spectrometric analysis. After adaptation, the usual techniques used in conventional 2D culture can be applied.

### 2.1. In Situ Characterization of Cells

The advantage of using techniques where the cells are embedded in the hydrogel allows for avoiding artifacts related to the dissociation of the hydrogel.

#### 2.1.1. Light Microscopy

Microscopy is particularly interesting in the characterization of hydrogels because it allows the structure of the construct to be preserved, as well as the cell–cell interactions. It allows access to the size and morphology of cells that could assemble into spheroids or in a native tissue organization. Phase-contrast microscopy allows for monitoring cell proliferation and growth over time without inducing toxicity [100,101,102]. However, because the cells are alive, the acquisition time should not be too long to avoid inducing cell death. This technique is only possible for optically transparent hydrogels. For example, the cell-ink bioink composed of alginate and cellulose nanofibril is opaque and does not track cells without prior fluorescent labeling or end-point histological analysis. 

Histological analysis requires sample preparation, including fixing, cutting, and staining [101,102,103]. The preparation steps for sectioning are very important. Dehydration for paraffin embedding tends to shrink the size of the sample and is therefore not be recommended for structural or organizational measurements [104]. In addition, if the hydrogel pores are not completely filled with paraffin, this will promote folding during sectioning and detachment of the sample from the section. However, the advantage of this technique is the possibility of having thin sections (up to 5 µm thick). In contrast, cryosection preserves the hydrogel structure, particularly with polyvinyl alcohol (PVA) and optimum cutting temperature (OCT) preparation. However, the sections are thicker, and more aspecific markings can be observed with a protein-based cryoprotectant solution [105]. Using resins favors the preservation of structures but makes it more difficult to perform histological stains [106]. Finally, it is possible to proceed directly to histological staining without cutting to visualize the cells on the surface of the hydrogel. Depending on the structures of interest, different stainings are available: Masson’s trichrome (TM) stains collagenous structures in blue (fibrosis, for example); hematoxylin (DNA) and eosin (proteins) illuminate viable zones in dark pink and dead zones in clear pink; and, finally, toluidine blue highlights the zones rich in RNA and DNA. Trypan blue is used to stain dead cells [107]. Quantification of chromatic staining can be difficult on thick samples, so the use of fluorescence microscopy is a good alternative.

#### 2.1.2. Fluorescence Microscopy

Fluorescence microscopy is used to label subcellular structures, such as the cytoskeleton (F-actin), mitochondria (MitoTracker), nuclei (Hoechst), or other types of organelles or proteins [108,109,110,111]. Standard immunofluorescence or biomarker labeling protocols can be applied to the hydrogel, although the times of the different labeling steps should be increased or even improved using mechanical agitation or a vacuum. Observation of the organization and viability of cells as a function of the position or shape of the hydrogel is only possible under microscopy. Using markers or antibodies coupled to fluorescent probes, it is possible to determine whether cells are dying (p-casp3), proliferating (KI67^+^ or DNA), entering in senescence (p16 or β-galactosidase), or in a hypoxic environment (HIF1-α, EF5, pimonidazole). Numerous fluorescence assays for dead/live cells are described in Table 2; however, the most commonly used combination of fluorochromes is calcein AM stain for esterase activity (live cells) and propidium iodide for permeable and therefore dead cells. It is possible to combine one of these two markers with Hoechst3342 or DAPI; however, this is not possible in all types of hydrogels, such as alginate, which shows strong auto-fluorescence from the UV channel. An easy-to-use marker for studying cell morphology is phalloidin labeling of F-actin, which is particularly interesting in models for studying mechanotransduction as a function of support stiffness, for example [107,112,113,114].

For high-resolution microscopy, confocal imaging is the reference method for studying cells embedded in the hydrogel. The disadvantage is the necessity to print thin film constructs on suitable substrates. Indeed, without a clearing technique, only 100 µm-thick constructs can be imaged. Furthermore, these hydrogels should preferably be printed on glass coverslips to favor high-resolution imaging. The risk is that the hydrogel may become detached; to mitigate this, the silanization of the coverslips allows the covalent bonding of the gel with its support. To limit the constraints of confocal imaging, other microscopy techniques have been developed, such as light sheet imaging. It is thus possible to image large objects without a physical section with limited phototoxicity [108,109,110,111].

#### 2.1.3. Electronic Microscopy

Electron microscopy provides nanoscale imaging, either scanning for the sample’s surface or transmission for the internal structures [102,115]. These techniques allow the study of cell–cell or cell–ECM interactions but also cell death. However, the sample preparation steps can change the structure of the sample.

#### 2.1.4. Colorimetric and Fluorimetric Methods

It is also possible to access the viability, proliferation, and metabolism of cells contained in hydrogels without dissociating them via colorimetric or fluorimetric methods. For this purpose, the use of prestoblue or alamar blue is an interesting solution because it is non-toxic for the cells, and it is possible to follow viability and proliferation over time. This test is based on reducing resurin to form resorufin, a red fluorescent compound. The test can be revealed by fluorimetry or absorbance reading [116,117,118]. Many tests exist to measure mitochondrial activity in cells, such as MTT, MTS, XTT, WTS, and CCK8, but MTT, for example, requires the dissolution of formazan crystals [112,119,120,121]. These tests are toxic to the cells and should be performed as an endpoint. All these techniques can be coupled with the measurement of lactate dehydrogenase release in the cell supernatant, revealing the membrane permeability of the cells [119]. Furthermore, background noise can be detected, so it is always necessary to make hydrogels without cells to manage this parameter. Generally, the same protocol as for the cells can be used; however, the incubation times may, in some cases, be slightly longer. Kits have been developed and adapted for 3D hydrogels, such as Celltiter-Glo 3D, which favors the penetration of reagents and has a better lytic capacity [101,116,122]. 

#### 2.1.5. Metabolic Fluxes Analysis

Cancer cell metabolism greatly influences tumor growth and resistance to anti-cancer treatments [123,124]. The organization of cancer cells into hydrogel allows for mimicking the heterogeneity that can be found in tumors in vivo, as the difference in access to nutrients and oxygen highlights the need to develop techniques to analyze the metabolism of cells in the hydrogel. Metabolic fluxes provide a detailed metric of the cellular metabolic state. For example, it is possible to place fluorodeoxyglucose ([18F]-FDG tracer) in the culture medium and monitor its localization and consumption in the hydrogel. This technique is particularly interesting, as it can be performed in animals in the pre-clinical study phase; however, the resolution is rather limited (1.5 mm on average) [120,125]. Glycolysis and mitochondrial respiration can be assessed by the Seahorse XF flux analyzer via the extracellular acidification rate (ECAR) or the oxygen consumption rate (OCR) [126,127,128]. However, this measure is global and does not consider heterogeneity in the sample. Furthermore, it is necessary to print small constructs with a large number of cells and sometimes to extend the measurement time.

The flux of extracellular metabolites can be determined by taking a small volume of the culture medium in contact with the spheroids at regular intervals (for example, every 4 h) [129,130]. The measurement of glucose and lactate, as well as glutamine and glutamate, can be measured using a clinical biochemistry analyzer or different kits. Extracellular fluxes were determined from the slopes of the fitted functions (mol/h/L) and by normalizing these results by the number of cells and the volume of the well. Finally, the intracellular metabolic flux can be determined using isotopes using NMR (nuclear magnetic resonance) [130,131] or GC–MS (gas chromatography–mass spectrometry) [129,132] and materials other than NMR (0.5 mg dry mass) [132]. From the cytoplasmic extracts, it is possible to determine the active amino acid synthesis pathways.

### 2.2. Characterization of Cells after Isolation or Lysis

#### 2.2.1. Molecular Biology

For many applications in tissue engineering, it is necessary to be able to extract DNA, mRNA, or protein in order to monitor different cellular parameters such, as differentiation or certain functions. It is also possible to determine proliferation and viability via the measurement of DNA concentration. This technique is interesting since it allows for knowing the number of cells per gel but also for normalizing the data obtained to the number of cells. However, it is critical to find the right technique to lyse the cells to recover the full amount of DNA. For example, for alginate gels, it is possible to use a commercial solution, the purelink genomic DNA mini kit [133]. For GelMA or agarose, the use of EDTA associated with proteases allows the recovery of cells from the gel in order to assay the DNA [113,114,134,135,136,137].

Conventional methods for 2D cell culture rely on two methods: either via the use of phenol/chloroform or with commercial kits using silica membranes in spin columns [138]. However, the inclusion of cells in hydrogels makes this step more difficult, and it presents more challenges that are technical. Indeed, the classical RNA extractions often do not allow for obtaining RNAs in sufficient quantity and/or quality for the subsequent performance of RTqPCR. Köster’s team conducted a study to investigate homogenization methods and RNA extraction techniques based on the most commonly used hydrogels (alginate, gelatin, and agarose) on hMSC cells [139]. For this purpose, four homogenization techniques are deployed. Regardless of the type of hydrogel, homogenization techniques using liquid nitrogen or a rotor stator should be excluded, as the yield of RNA is very low. In contrast, the amount of RNA is much higher for techniques using the micro-homogenizer or enzymatic/chemical digestion. The technique of frozen liquid nitrogen crushed by an electric crusher seems to be relevant for GelMA-type homogenization [113,134]. For extraction, Köster’s team shows that conventional commercial kits using silica membranes in spin columns do not provide a correct RNA yield for hMSC in alginate, gelatin, and agarose hydrogels [139]. However, other teams obtain satisfactory results with agarose-based or alginate hydrogels [114,140]. Hot phenol (HP), TRIzol (TR), cetyltrimethylammonium bromide (CTAB), and LiCl (LC) techniques have a better RNA yield, but the LiCL technique gives poor PCR results (e.g., dominating additional band, PCR product with incorrect size or no PCR product). For the same reasons, the TRIzol technique is not adapted for alginate gels for Köster’s team but it is for Ewa-Choy’s or Sbrana’s Teams [118,133]. Hot phenol and CTAB seem to be the most suitable techniques; hot phenol gives the best RNA yield, and CTAB gives the best RNA quality (equivalent to 2D culture) and low endpoint Ct values ~20.

#### 2.2.2. Flow Cytometry

It can also be interesting to isolate cells to either promote cell expansion or analysis using flow cytometry, as this allows many cells to be analyzed very quickly. For this purpose, enzymatic degradation is possible for matrices derived from natural products, such as collagenase for GelMA or collagen hydrogels, hyaluronidase for hyaluronic acid-based gels, or alginate lyase for alginate hydrogels. Some materials can also be degraded by physical techniques, such as photo-degradation [141]. This step is critical because a too-prolonged enzyme treatment can induce significant cell death or even alter the membrane receptors. The limitation of this technique also lies in the fact that a large hydrogel is required to recover the necessary number of cells after degradation [142]. Then, standard labeling protocols such as 2D culture can be used. Flow cytometry allows quantitative measurements of many parameters simultaneously, such as viability, proliferation, cell cycle, and uptake of anti-cancer agents. As for microscopy, live/dead tests based on calcein AM and ethidium are the most commonly used, with propidium iodide or BrdU for the cell cycle. It is also interesting to use this technique to identify subpopulations or maintenance of a phenotype, such as chronic lymphocytic leukemia cells on CD5^+^CD19^+^IgM^+^ markers [118]. The disadvantage of flow cytometry compared to microscopy is the loss of spatial information. To compensate for this, Beaumont’s team developed a protocol based on the diffusion gradient of Hoechst 33342, which makes it possible to discriminate between internal and peripheral cells according to the intensity of Hoechst [143].

**Table 3 ijms-23-03432-t003:** Characterization technology of bioprinted constructs. Value of 3D bioprinting for cancer modelling. + for pros and − for cons.

Methods		Description	Pros and Cons	Markers	REF
Microscopy					
Light	Phase contrast	Monitoring of proliferation and morphology of cells	+: • Nondestructive • No markers are added • Low cost • Easy with transparent gels (GelMA, matrigel) −: • No possibility to identify subcellular structures • Difficult with opaque or non-transparent gels (e.g.,: alginate with nanocellulose)	Not suitable	[100,101,102]
	Bright field	The transmission of light is more or less attenuated depending on the density or marking of the sample	+: • Suitable for large samples −: • Requires histological staining • Preparation of sample • Quantification of thick sample	Hematoxylin–eosin Masson’s trichrome Trypan blue	[101,102,103]
Fluorescence	LSM Epifluorescence Confocal	The use of a fluorescent marker is necessary to highlight a subcellular structure; possibility of monitoring structures over time (if vital markers)	+: • Monitoring of many possible structures −: • Requires cutting for oversized constructions for epi and confocal microscopy • Need to fix for certain markers • Important autofluorescence for chitosan or alginate/cellulose hydrogels in UV	Live/dead staining Or calcein AM/propidium iodide Or ethidium homodimer Active-caspase3/7 green Hoechst 33342 HIF1-α, Ki67	[108,109,110,111,144]
Electronic	Scanning	Surface is scanned with a beam of electrons, emitted signal provides images	+: • High resolution −: • The preparation procedure is tedious • Frequent preparation artifacts (collapse)	Not suitable	[102]
	Transmission	The part of beam of electrons is transmitted into specimens allowed to obtain images		Not suitable	[102,115]
Flow cytometry					
Flow cytometry		Analysis of physical parameters (size and granularity) for each cell but also the level of fluorescence	+: • Quantitative analysis −: • Disaggregation can be a problem • Necessity to have a large cell number due to loss of cells during dissociation	7-AAD CFSE	[102,139]
Spectroscopy					
Spectrometry or fluorimetry		Production or utilization of a fluorescent or chromatic compound	+: • Well-described for 2D culture and frequently used • Can be used for kinetic monitoring −: • Ensure that the efficiency is adapted for 3D	ACP, LDH, prestoblue, alamar blue, DNA content	[112,119,120,121]
Molecular biology					
RTqPCR Western blot		Quantification of gene expression at mRNA or protein level	+: • Quantitative analysis • Easier by using the enzymatic method on natural inks (e.g., collagenase for GelMA or ColMA, hyaluronidase for hyaluronic acid) −: •Adaptation of the homogenization and extraction protocol to obtain an adequate quantity and quality of RNA/proteins for analyses	Bax/Bcl2 HIF1-α, Ki67	[103,115,118]
Metabolism					
GC–MS (Gas chromatography–mass spectrometry)		Detection of molecules of interest according to their mass/charge ratio after ionization	+: • Considerably less cellular material compared to NMR, high sensitivity, −: • Use of radioisotopes, complex sample preparation, high cost	^13^C-Glucose	[129,132]
NMR (nuclear magnetic resonance) spectroscopy		Determination of the composition of a sample by applying a magnetic field via the orientation of the nuclear spins of the atoms	+: • High reproducibility, sample can be analyzed directly, low cost −: • Use of radioisotopes, low sensitivity		[130,131]
PET scan (positron emission tomography)		Injection of a radiographic tracer and monitoring by imaging to detect localization of [^18^F]FDG	+: • Classically used in medicine, monitoring over time −: • Low resolution (1.5 mm)	[^18^F]FDG	[120,125]
Seahorse		Quantification of the oxygen consumption rate (OCR) and the extracellular acidification rate (ECAR)	+: • High sensitivity (from 5000 cells, theoretically), possibility to test many conditions in parallel −: • Difficulties in normalizing results, limited number of injections, limited sample thickness	Not suitable	[126,128]

7-AAD: 7-ADDminoactinomycin; [18F]-FDG: 18F-2-Fluor-2-deoxy-D-glucose; ACP: acid phosphatase assay; CFSE: carboxyfluorescein succinimidyl ester; CTV: celltraceviolet; MTS: 3-(4,5-dimethylthiazol-2-yl)-5-(3-carboxymethoxyphenyl)-2-(4-sulfophenyl)-2H-tetrazolium; MTT: 3-(4,5-dimethylthiazol-2-yl)-2,5-diphenyltetrazolium bromide; pNPP: p-nitrophenyl phosphate; PET: positron emission tomography; WST: water-soluble tetrazolium; XTT: 2,3-bis-(2-methoxy-4-nitro-5-sulfophenyl)-2H-tetrazolium-5-carboxanilide.

Actual models for cancer study range from in vitro traditional 2D cultures to in vivo models; most of the time, the complexity of the model goes hand in hand with the complexity of assaying the subsequent metabolism [145]. Three-dimensional bioprinting allows for adding high-complexity tissue modeling in a relatively user-friendly technology (Figure 4). Compared to the widely used organoid approach, 3D bioprinting allows, in an automated way, the creation of complex 3D structures with the precise and reproducible deposition of cells and matrices.

Bioprinting is, therefore, an innovative approach to mimic the in vivo microenvironment of cancer cells as closely as possible (Figure 5). This has the advantage of producing more viable results that are closer to in vivo results, such as cell–cell or cell–ECM, or resistance to treatment as a function of the microenvironment. One could also imagine mimicking the tumor microenvironment for each patient (personalized medicine) or for a cohort (biobanks) to test their responses and resistance to the different therapeutic lines. New bioprinting methods have also made it possible to obtain a greater number of cancer stem cells, cells that are particularly difficult to maintain in vitro and incriminated in cancer relapse.

### 2.3. Recapitulate Cancer’s Relation to the Microenvironment

#### 2.3.1. Cells–ECM Interaction

For a long time, the study of cancers was solely based on the precise genetic, metabolic, and phenotypic analysis of single tumor cells, with tumor stroma being totally ignored [146]. Recently, there has been strong evidence of stroma–tumor interactions related to tumor progression [147]. This cancer stroma is a complex framework of supportive tissue composed of the extracellular matrix (ECM), cells (such as fibroblasts and adipocytes), inflammatory and immune cells, and a specific vascularization. Thus, there are complex interactions between the stroma and the cancer cells: cancer cells can modify their stroma, and stroma can support tumor progression. 

Adipocytes are a main component of the human body and are thus in the vicinity when tumorigenic events take place [148]. Complex crosstalk is then set up, in which phenotypical and functional modifications of both tumor cells and adipocytes occur. Adipocytes release fatty acids that can be oxidized in cancer cell mitochondria and thus provide energy through ATP in times of metabolic need [123]. In breast cancer, aberrant adipocytes called cancer-associated adipocytes (CAA) are known to promote the invasion and metastasis of breast cancer, in particular through the secretion of adipocytokines in the invasive front of the tumor [149]. Horder et al. bioprinted a breast cancer model with adipose-derived stromal cells (ADSC) [64]. ADSCs were differentiated into adipocytes within the hyaluronic acid gel and allowed the remodeling of the ECM with increased collagens I and IV and fibronectin expression, demonstrating the important interactions between cancer cells and adipose tissue.

Cancer-associated fibroblasts are another key component of the tumor microenvironment, notably through their capacity to remodel the extracellular matrix but also through direct cellular interactions via paracrine signals (exosomes, metabolites, and cytokines) with cancer and immune cells [150,151]. In a recent paper, Hanley et al. showed that CAF could be a potential target to overcome resistance to anti-PD1/PD-L1 and CTLA-4 immunotherapy [152]. Mondal et al. printed non-small cell lung cancer (NSCLC) patient-derived xenograft (PDX) cells and lung CAFs that allowed high viability and efficient crosstalk [153].

Spheroids have long been used to complexify tumor models, but despite their 3D structure, they are not sufficient to recapitulate the complexity of the microenvironment, notably due to the lack of multiple cell types and vascularization. Three-dimensional bioprinting allows for recapitulating the complexity of the tumor microenvironment, particularly through the precise deposition of several cell types, the ability to vary the type of matrix, and the ability to precisely set up vascularisation networks [154,155]. As reported by Samadian et al., ECM components and cells have a crucial role in the progression and spread of cancers, and 3D bioprinting allows for mimicking the tumor microenvironment at physical, cellular, and molecular levels [156]. The possibility of making sacrificial templates using sacrificial materials (e.g., pluronics F-127) allows the setting up of vessel-like structures that can be cellularized and perfused, improving nutrient availability [157]. Different strategies can be used for the printing of a vascular network that is recapitulated by Richards et al., but extrusion-based bioprinting is quite capable of printing complex networks (for review, see [158]). 

#### 2.3.2. Neoangiogenesis

Angiogenesis is a normal mechanism by which new blood vessels can be generated. Angiogenesis is made up of different stages, including the degradation of the matrix via proteases and the migration and proliferation of endothelial cells to form new tubes that are anastomosed with pre-existing ones [159]. In a normal state, angiogenesis is mainly regulated by hypoxia, in particular through the hypoxia-inducible transcription factor (HIF) family [160]. To allow tumor growth, cancer cells will stimulate endothelial cells activity by releasing many soluble factors, such as EGF, FGF, and VEGF. Tumor-endothelial interactions are also essential in metastasis processes. 

Three-dimensional bioprinting allows for studying the mechanisms at the origin of neoangiogenesis. As reported by Zervantonakis et al., 3D breast adenocarcinoma bioprinted models associated with microfluidics can recapitulate changes in the endothelial barrier caused by tumor–endothelial cells interactions and model the process of intravasation [161]. In a model of lung carcinoma, 3D bioprinting of a vascularized tissue allowed for exploring the molecular mechanisms of metastasis by using a gradient of angiogenic factors, such as EGF and VEGF, in printed programmable release capsules [162].

#### 2.3.3. Migration and Invasion

Metastases are secondary cancers that originate from the migration and invasion of cells from primary cancer, and their occurrence is the main cause of cancer-related deaths (~90%) [163]. Understanding the mechanisms involved in the genesis of metastasis remains a major challenge in the fight against cancer. Currently, the 3D method classically used is the Boyden chamber, where cells will migrate through a physical barrier containing pores (migration) or combined with a protein coating, often the Matrigel, mimicking the ECM (invasion) [164]. However, this technique is often performed as an endpoint and does not allow screening with many conditions. Moreover, the Matrigel is not very reproducible from one batch to another in terms of composition and stiffness.

In this sense, the team of Jung et al. developed a bioprinting platform to study the migration or invasion, i.e., the migration through the extracellular matrix of tumor cells, with a drop-on-demand inkjet 3D bioprinter [165]. This technique allows for analyzing the simultaneous migration of 96 conditions in parallel with very reproducible results in hydrogels of varying stiffness, for example.

#### 2.3.4. Enrichment in Cancer Stem Cells

A tumor comprises many differentiated cancer cells and a few cancer stem cells (<0.001% of tumor cells). Relapse of some types of cancer may occur due to the presence of cancer stem cells (CSCs) that are resistant to various anti-cancer therapies (chemo-, targeted, or radiotherapy) or low pH, oxygen, or glucose content [166] (Figure 6). These cells have the capacity for self-renewal, low cycling, and the ability to differentiate. The CSCs are particularly difficult to study in vitro as they are very poorly represented and do not have a single stemness-specific marker but a combination that does not strictly define this population. However, they have a phenotype that can drift under conventional culture conditions. Until now, the majority of methods used to isolate this CSC population were based on low-adherence and low-oxygen culture conditions, as well as stimulation by chemotherapy [167,168,169]. Unfortunately, these techniques were time-consuming and did not always allow for the recovery of many cells. Recently, three-dimensional culture systems have been developed to enable conventional two-dimensional monolayers to mimic the in vivo microenvironment as closely as possible [168].

The use of bioprinted cells in hydrogels may overcome these limitations, and the ability to mimic the native elastic environment of the cells may help to maintain stemness. The Suzuka team demonstrated that the seeding of cells in a double-PEG hydrogel network allowed for rapid reprogramming of human differentiated tumor cells into CSCs within 24 h of culture, for six human cancer cell lines (affecting brain, lung, uterine–cervix, colon, bladder, and synovium organs) or with brain cancer cells resected from patients with glioblastoma [170]. In all these conditions, upregulation of cancer stem cell marker genes is observed (Oct3/4, Nanog, and Sox 2). Similarly, including cancer cells in alginate beads, alginate/chitosan, chitosan/hyaluronic acid, or collagen hydrogel increases the proportion of CSC with an upregulation of stemness genes for glioblastoma, breast, hepatocellular, and prostate cancer [36,171,172,173,174,175,176]. Similar results can be obtained with fibrous materials made of PCL [177].

### 2.4. Mechanical Environment

#### 2.4.1. Mechanotransduction

It has now been well-known for many years that cellular metabolism cannot be reduced to the functioning of an isolated cell. Cells grow and interact with their environment, notably via chemical and physical factors that can drive their fate. This mechanism of sensing, integrating, and responding to external signals is widespread in almost all living organisms. Chemical interactions mediated by soluble factors or cell–cell interactions have been extensively studied in the past; however, cell interactions with their environment and notably with the extracellular matrix (ECM) cannot be reduced to chemical stimuli. In recent years, physical cues have proved to be major regulators of the cell response to external stimuli, including the ability to sense external applied forces, rigidity, topography, and orientation [178,179,180]. The mechanism by which these external physical stimuli are detected, transmitted to the cell, and converted into biochemical information is called mechanotransduction [181]. The detection of external stimuli, also called mechanosensing, depends on the nature of the signal and is particularly mediated through focal adhesion complexes (FAs) (composed of multiple mechanosensors, such as talin and vinculin), adherens junctions, and mechanically activated channels (e.g., Piezo) (for review, see [182]). The microenvironment can induce different physical and mechanical stresses on tumor cells. The cell can be subjected to three different types of mechanical stress: (i) tensile stress, related to the contraction of actomyosin during the stiffening of the ECM; (ii) compressive stress, due to the anarchic proliferation of cells in a confined space during tumor growth phases; and (iii) shear stress with blood and interstitial fluid pressure. Among the physical determinants of mechanotransduction, stiffness has proved to be a major regulator of cell metabolism. Stiffness is a term used to describe the force necessary to obtain the deformation of a structure [183]. In cell biology, the stiffness of a tissue is mainly derived from ECM composition and thus the proportion of its components that are mainly represented by fibrous-forming proteins, e.g., collagens, elastin, and fibronectin (for review, see [184]). Among them, hyaluronan acid and collagens are the main determinants of ECM stiffness. Information derived from ECM stiffness can then be converted by the cells and influence their fate, particularly through changes in their metabolism [185]. One remarkable feature of cancer cells is the capacity to change their metabolism to adapt to the harsh conditions of their specific tumor environment and adapt to the aberrant signaling induced by oncogenes or tumor suppressors [186]. Thus, there is a complex dialogue between the cancer cells and the tumor microenvironment as the cells can change their composition and stiffness, and in turn, the change in stiffness can lead to changes in cancer cell metabolism.

#### 2.4.2. The Link between Extracellular Stiffness and Cell Metabolism

The modification of the rigidity of the microenvironment modifies the intracellular tension, which in turn will retroact on the rigidity of the tissue. Indeed, on stiff substrates, the integrin clusters are activated and, via their binding to talin, promote actin polymerization. This generates significant intracellular tension, which will allow the recruitment of proteins, such as vinculin, which will stabilize talin and potentiate the activation of FAK (focal adhesion kinase) and Rho/Rho kinase (ROCK), leading to the maturation of focal adhesions and the assembly of stress fibers [182,187,188]. Once this external mechanical stimulus is propagated to the cytoskeleton, it will induce the structural modifications of membrane proteins and their translocation to the nucleus. The perception of those external physical cues and their transmission to the nucleus are determinants for proper cell function and metabolism.

##### Amino Acid, Glucose, and Lipid Metabolism

The effects of ECM stiffness on glucose metabolism are mediated by several pathways, of which the YAP/TAZ and FAs are predominant (see Figure 7). As reported by Ge et al., there are several other coupling pathways inducing modifications of nutrient metabolism, which will not be described in the following (for review, see [185]). The main glucose metabolism changes induced by extracellular stiffness are represented by changes in glucose uptake by GLUT transporters, regulation of glycolysis, regulation of the pentose phosphate pathway through the PI3K-AKT pathway, and glycogen metabolism through AMPK [189,190,191,192,193]. Amino acid metabolism is also affected by extracellular stiffness, in particular the modulation of proline synthesis, which is mediated by Kindlin-2 [194,195]. YAP/TAZ, through mTORC1, will also potentiate amino acid uptake, and activation of the PI3K/AKT pathway appears to upregulate SLC6A19 protein, a promoter of amino acid uptake [196,197]. Finally, lipids are also regulated by extracellular stiffness, in particular through SREBP1 and SREBP2, which are the main regulators of cholesterol and fatty acid synthesis [198]. Furthermore, stiffness will also regulate some cell membrane lipid receptors, such as CD36 and LDLr, as examples [199,200].

##### Nucleus and Cell Cycle

Matrix stiffness also regulates cell growth and protein synthesis. As reported by Tilghman et al., cells on soft substrate ECM (from 150 Pa to 300 Pa) had longer cell cycles and were metabolically less active than cells growing on high-rigidity substrates (superior to 10,000 Pa) [201]. In this study, cells from human lung carcinoma (A549) show an increase in G1 phase and low ATP levels and protein synthesis when cultured in soft substrates, which may be explained by the entry into a stage of tumor dormancy. However, these results are not transposable to all cancer cells, some being rigidity dependent, i.e., characterized by a proportional increase in growth with stiffness, and “rigidity independent” [202]. This regulation of genome expression by mechanotransduction is called “nuclear mechanotransduction,” and such transduction is possible through either the cytoskeleton network that bridges the cell membrane to the nucleus or by activating secondary messengers in the cytoplasm that will secondarily translocate to the nucleus (for review, see [203]).

##### Mitochondria

Mitochondrial morphology is also impacted by extracellular stiffness. In a 2017 study by Lyra-Leite et al., ECM elasticity was proven a major regulator of mitochondrial metabolism in cardiac tissues [204]. Mitochondrial respiration was highly dependent on substrate stiffness, with higher basal, maximal, and spare respiratory capacity (SRC) in highly elastic modulus culture conditions. SRC is a determining parameter of mitochondrial adaptation; ECM thus seems essential in regulating mitochondrial metabolism [128]. In bovine vascular smooth muscle cells, changes in substrate stiffness were responsible for variations in mitochondrial cluster size and TMRM intensity [205]. The mechanism through which stiffness impacts mitochondrial metabolism is still lacking understatement; however, recent studies have shown that high-stiffness ECM promotes mitochondrial fusion and, at the same time, impedes fission, in particular, through DRP1 inhibition [206]. This is also in agreement with the study of Lyra-Leite et al., since a state of fusion is known to be responsible for better mitochondrial function [204,207]. 

## 3. The Link between Stiffness, Cancer, and Resistance to Anticancer Therapies

During tumorigenesis, ECM undergoes significant changes [208]. Peritumoral ECM accumulation results in an intense fibrotic response, also called desmoplasia, and increased stiffness. As reported above, changes in stiffness will induce changes in cancer cell metabolism. Increased stiffness will thus promote a tumor vascular phenotype but also epithelial–mesenchymal plasticity and tumor metastasis in different tumor models [209,210,211]. Matrix stiffening induces modifications of tumor vascularization, increasing angiogenesis, neovessels branching, and invasion [211]. Epithelial–mesenchymal transition (EMT) also seems to be controlled by matrix stiffness, as reported by Rice et al. Fibrotic rigidity that can be found in pancreatic cancer promotes EMT elements towards a mesenchymal phenotype, leading to paclitaxel resistance [212]. This EMT transition has been elucidated in breast cancer by Fattet et al. via a mechanoresponsive EPHA2/LYN complex that promotes breast cancer invasion and metastasis [209]. Microenvironment rigidity also seems to increase resistance to conventional cancer chemotherapies and targeted therapies in different cancer models [212,213,214]. 

It is therefore essential to have controlled stiffness models to properly model cancer resistance to therapies. Three-dimensional printing allows the large choice of bioinks and the different extrusion and crosslinking methods to precisely control the stiffness of the printed construct. As an example, alginate bioinks can be tuned to have specific mechanical properties (Young’s modulus and degradation rate) that will drive the differentiation of MSCs towards osteogenesis or adipogenesis [215]. In 2021, Kuzucu et al. were able to mimic the graded stiffness architecture that can be found in tissues by using bioinks composed of carboxylated agarose [216]. Regarding cancer research, Monferrer et al. showed that mixing GelMA and different concentrations of AlgMa allowed the creation of gradients of stiffness that highlighted the role of intercellular space stiffness on the clinical behavior of neuroblastoma [217].

### 3D Bioprinting for Drug Delivery and Screening

In developing new drugs, in vitro studies are essential before moving on to preclinical studies. Even if preliminary high-throughput drug screening is nowadays mostly completed on 2D cultures, simple 3D-bioprinted models could increase the relevance of this first screening while allowing for the speed and reproducibility required at this stage, which would improve the relevance of target candidates [218]. Once drug candidates have been selected, 3D bioprinting allows for fabricating more complex pathologic models (organ-on-a-chip), narrowing the gap between initial in vitro studies and final animal testing (for review, see [219]).

Many models have already been developed; Mao et al. used patient-derived cholangiocarcinoma cells to print a 3D-bioprinted construct to test sorafenib, cisplatin, and 5-fluorouracil resistance [45]. They showed that bioprinted cholangiocarcinoma cells showed stem-like properties and high resistance to all of those drugs compared to the 2D culture. Lee et al. used a fibrin-based bioink to allow glioblastoma cells to form spheroids within the construct and with an altered response to novel glioblastoma treatment methods [68]. Breast cancer 3D bioprinted models (MCF10A and MDA-MB-231 cells) have been extensively used (for review, see [220]). Breast cancer cell resistance is altered in 3D models compared to 2D models with a notably increased resistance to tamoxifen. 

Apart from drug-testing platforms, 3D-printed biomaterials may also be used as drug-delivery vehicles [221]. Alginate-based drug-delivery hydrogels have been used for breast, prostate, and colon cancer to enhance efficiency, in particular, through local delivery, sustained action, and enhanced uptake activity [40]. In skin, 3D bioprinting allows for creating personalized drug-loaded patches to deliver salicylic acid for acne treatment [222]. Bioprinting also allows the coating of microneedles to allow precise cisplatin delivery in skin cancer.

The emergence of new technologies, such as direct-volumetric drop-on-demand (DVDOD) bioprinting, will allow high-throughput drug-delivery models that will use bioprinting as a personalized drug-delivery platform [223].

## 4. Conclusions

Three-dimensional bioprinting in the past few years has made outstanding progress to become a major translational tool. Bioprinting has allowed for the fabrication of many constructs for tissue engineering and spheroid/organoid models or complex constructs for oncology. To find the most appropriate treatment for each patient, it is possible to bioprint or generate live tumor spheroids [224]. It will then be possible to find the right combination of treatments and the required dose for each patient in a personalized and precise manner. The generation of these models from patient biopsies makes them an optimal preclinical model for cancer drug screening. The creation of a biobank for each type of cancer in a 3D model will make it possible to obtain results that are more reliable and closer to clinical data than the results previously obtained from 2D models, where treatment doses are often underestimated. However, the need to print and use the constructs extemporaneously without the possibility to preserve the printed constructs remains a major limitation. A recent publication shows that Ravanbakhsh’s team has developed a new method of cryoprinting and cryopreservation for cell-laden tissue constructs to overcome this problem [225]. 3D biopriting will be a major tool for cancer research in the years to come due to its ability to overcome the limitations of 2D cell culture by adding the complexity of the microenvironment in a reproducible and repeatable manner that will allow for the quick modelling of in vitro personalized tumor models. Through the precise deposition of cells and biomaterials, the complexity of the tumor niche can be reproduced: necrotic/hypoxic core, gradients of stiffnesses, perfusion, microenvironment cells, etc. Drug testing will be thus greatly facilitated, and biopriting will facilitate the choice of pertinent candidate anticancer therapies. Moreover, it will allow the reduction of the use of animal testing, which is nowadays a major concern, as the recent referendum on the ban on the use of animal experiments in Switzerland shows. 

The different technologies presented in this review allow us to re-create living tissues with ECM, vascularization, physical constraints, and metabolic activity, reducing the gap with in vivo studies. However, more steps need to be assessed to spread 3D bioprinting in all research teams, such as (i) the creation of standard printing guidelines, (ii) global harmonization in bioink formulations, (iii) the production of novel biomaterials with enhanced biological and physical properties, (iv) the improvement of post-printing processes and maturation of the printed construct, and (iv) the development of biological tests that can be conducted within the printed construct. The maturation of the 3D construct, also called 4D bioprinting (as it adds the time dimension), is one of the major factors in the years to come to allow the post-maturation of the 3D construct over time thanks to an external stimulus [226].

In this review, we highlight some major points in the bioprinting process, underlining the pros and cons of the different bioprinting technologies, bioinks, and crosslinking parameters to enable cancer researchers to make informed choices that will allow them to easily implement bioprinting in their laboratories.

## Figures and Tables

**Figure 1 ijms-23-03432-f001:**
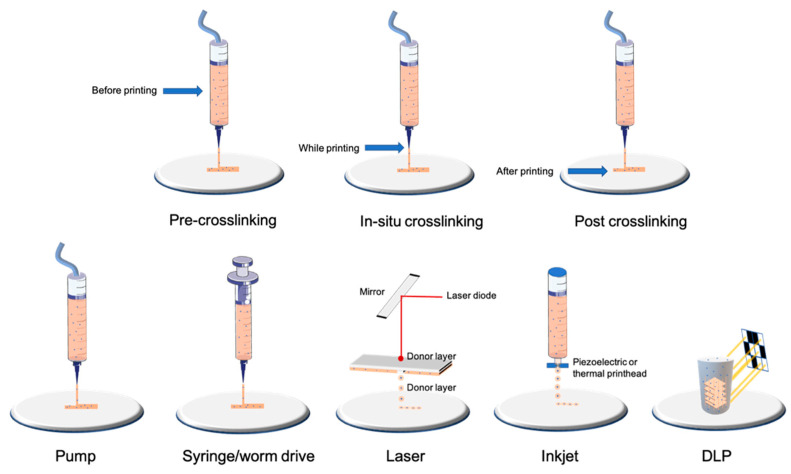
Examples of bioprinting and crosslinking technologies.

**Figure 2 ijms-23-03432-f002:**
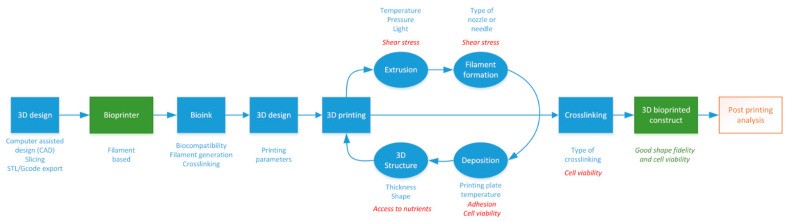
Flow chart proposal for the implementation of 3D filament-based bioprinting. Blue—key technical steps; red—influence of technical choices on cell status.

**Figure 3 ijms-23-03432-f003:**
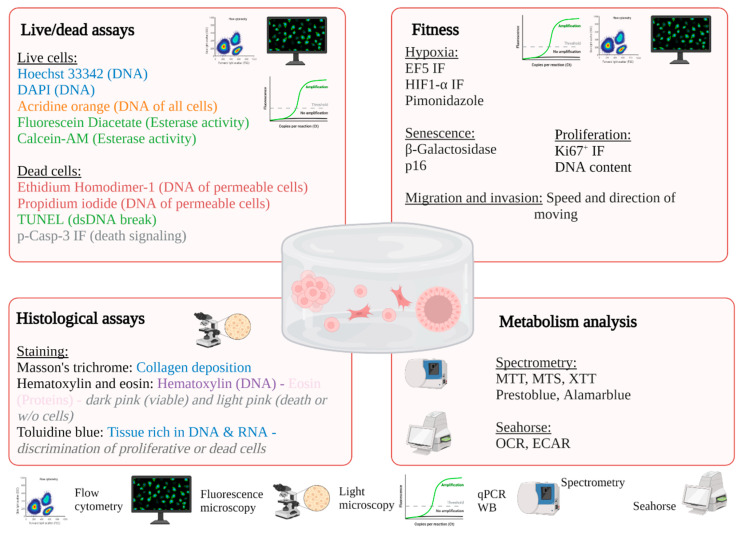
Examples of possible analysis of 3D bioprinted constructs.

**Figure 4 ijms-23-03432-f004:**
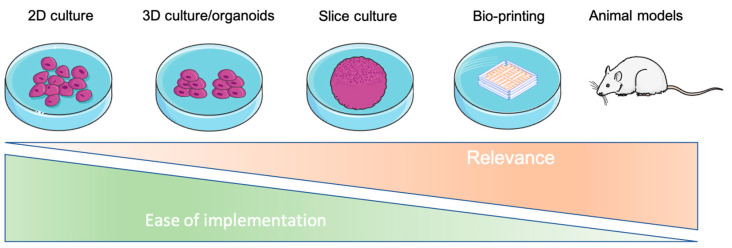
Relevance and ease of implementation of different research models.

**Figure 5 ijms-23-03432-f005:**
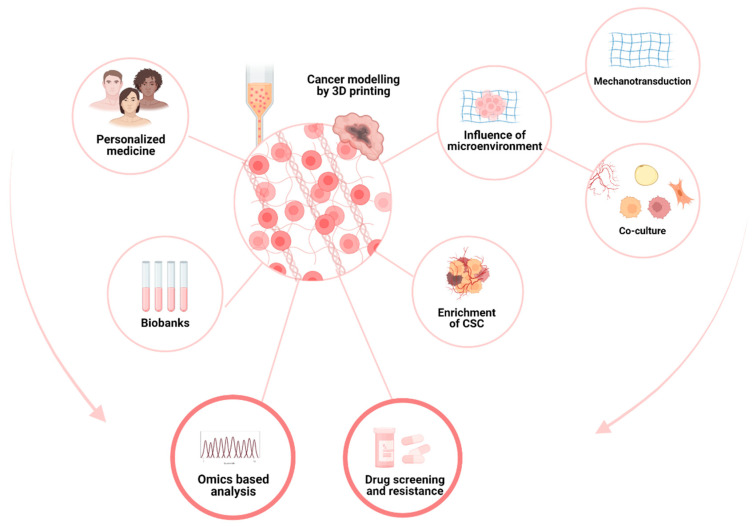
The value of bioprinting for oncology research.

**Figure 6 ijms-23-03432-f006:**
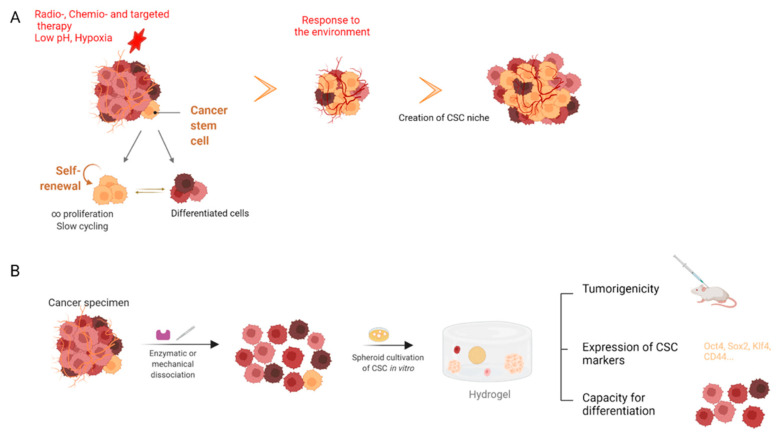
Cancer stem cell (CSC) and microenvironment. (**A**) A tumor is composed of heterogeneous cells, including a small fraction of cancer stem cells (in yellow). These cells are distinguished by their low cycling and their ability to self-renew and differentiate. When the tumor is exposed to treatments or a hypoxic or low-nutrient environment, these cells will resist and survive in a niche that is adapted to them. (**B**) A large number of CSCs is of interest to test an effective personalized treatment for each patient. For this purpose, a tumor sample must be dissociated by enzymatic and/or mechanical treatment and then cultured in a 3D environment to promote the formation of spheroids in the hydrogel. These cells then show the capacity of tumorigenicity (tumor formation in vivo); overexpression of stemness markers, such as Oct4 and Sox2; and, finally, the capacity to differentiate.

**Figure 7 ijms-23-03432-f007:**
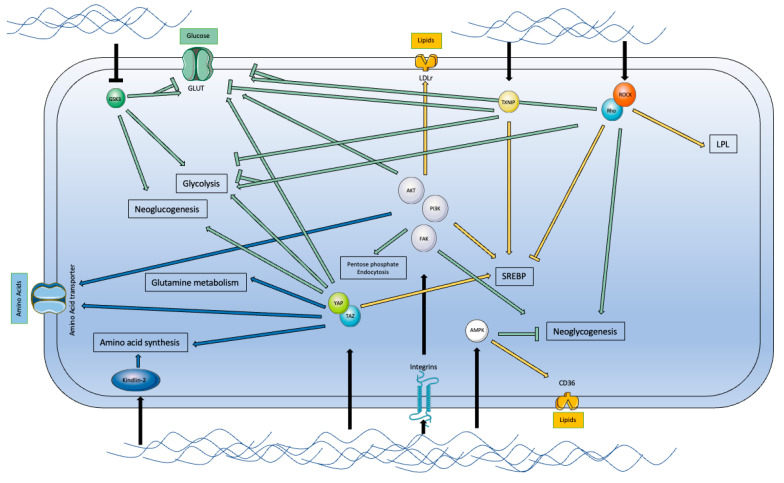
Main mechanotransduction pathways. In the green, the influence of mechanotransduction in glucose metabolism; in blue, the influence in amino acid metabolism; in yellow, the influence in lipid metabolism. GSK3: glycogen synthase kinase-3, GLUT: glucose transporter, LDLr: low-density lipoprotein receptor, TXNIP: Thioredoxin interacting protein, ROCK: Rho-associated protein kinase, LPL: lipoprotein lipase, PI3K: Phosphoinositide 3-kinase, FAK: focal adhesion kinase, YAP: Yes-associated protein, AMPK: AMP-activated protein kinase, SREBP: Sterol regulatory element-binding proteins, CD36: cluster of differentiation 36, also known as platelet glycoprotein 4, fatty acid translocase (FAT).

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
