# Peer review of "Current Advances in 3D Bioprinting for Cancer Modeling and Personalized Medicine"

_ijms, 2022, doi:10.3390/ijms23073432_

Round 1

Reviewer 1 Report

In the manuscript by Germain et al., the 3D-bioprinting technologies for in vitro cancer models to study cells grown in hydrogels were reviewed. The fundamentals of bioinks and cell behavior characterization are summarized, as well as some relevant applications of bioprinted cancer models were discussed in a broad field of view. The logic and writing are clear. The reference is update to date. The review can provide good information for the new learner in the field. There are several issues that can be addressed to improve the manuscript.

  1. The authors catalyze printing techniques into “droplet, filament, and plane-based printing”. The prevailing volumetric printing can also be discussed.
  2. UV crosslinking of GelMA is a typical chemical crosslinking instead of a physical method.
  3. The perspective of using 3D printing for cancer models can be expanded a bit.

Other minor comments:

  1. Page 2, line 82, “50 micrometer” is incorrectly shown.
  2. More updated reference in the 3D printing techniques and Many statements needs a reference. For example, “Even if the droplet generation (surface tension) and breaking combined with the force with which it will be projected onto the printing plate can reduce cell viability, drop-based approaches  allow  higher  cell  viability  than  filament-based ones(>85%).”

Author Response

We would first like to thank you for your careful review.

In regard with the points you have raised:

  1. The authors catalyze printing techniques into “droplet, filament, and plane-based printing”. The prevailing volumetric printing can also be discussed.

We thank you for your comment, we have added a short paragraph about volumetric bioprinting in the plane-based paragraph. We have added specific references to volumetric printing in the same paragraph.

  1. UV crosslinking of GelMA is a typical chemical crosslinking instead of a physical method.

We thank you for your comment, modifications have been made on the table 2.

  1. The perspective of using 3D printing for cancer models can be expanded a bit.

We thank you for your comment, a paragraph on 3D printing for cancer models has been added in the conclusion.

“3D biopriting will be a major tool for cancer research in the years to come due to its ability to overcome limitations of 2D cell culture by adding the complexity of the microenvironment in a reproducible and repeatable manner that will allow  to quickly model in vitro personalized tumor models. Through the precise deposition of cells and biomaterials, the complexity of the tumor niche can be reproduced: necrotic/hypoxic core, gradients of stiffnesses, perfusion, microenvironment cells…Drug testing will be thus greatly facilitated and biopriting will facilitate the choice of pertinent candidate anticacer therapies. Moreover, it allows to reduce the use of animal testing which is nowadays a major concern, as the recent referendum on the ban on the use of animal experiments in Switzerland shows.”

Other minor comments:

  1. Page 2, line 82, “50 micrometer” is incorrectly shown.

We thank you for your comment and have corrected it in the manuscript

  1. More updated reference in the 3D printing techniques and Many statements needs a reference. For example, “Even if the droplet generation (surface tension) and breaking combined with the force with which it will be projected onto the printing plate can reduce cell viability, drop-based approaches allow  higher  cell  viability  than  filament-based ones(>85%).”

We thank you for your comment and have added references to the concerned paragraph.

You will find attached the corrected manuscipt.

Regards,

Nicolas Germain

Reviewer 2 Report

This focused review describes the recent advances and progress in the field of 3D bioprinting for cancer modeling and medicine. The work has been prepared in a comprehensive format and contains valuable data for experts, scientists, and students. The work has a strong potential to be published here. However, I do have some suggestions, as listed below:

1) In the bibliography section: The last sentence of the first paragraph needs appropriate articles to be considered, such as Nature materials, 20(5), 593-605 (2021), Biosensors and Bioelectronics 177, 112971 (2021), etc. 

2) There must be more stress on the sample to results duration in various systems. Such quantitative results/data can be listed in a table.

Author Response

We would first like to thank you for your careful review.

In regard with the points you have raised:

  • In the bibliography section: The last sentence of the first paragraph needs appropriate articles to be considered, such as Nature materials, 20(5), 593-605 (2021), Biosensors and Bioelectronics 177, 112971 (2021), etc. 

Thank you for this comment, we have added relevant publications at the end of this first paragraph:

Tino, R. et al. COVID-19 and the role of 3D printing in medicine. 3d Print Medicine 6, 11 (2020).

Choong, Y. Y. C. et al. The global rise of 3D printing during the COVID-19 pandemic. Nat Rev Mater 5, 637–639 (2020).

  • There must be more stress on the sample to results duration in various systems. Such quantitative results/data can be listed in a table.

We thank you for this remark however, it is difficult to give precise figures for the time between the collection of the cell sample and the bioprinted construct as this depends on too many parameters (size of the construct, technology used, polymerisation etc.). For this reason, the speed of the technology is only specified in semi-quantitative terms (in table 1).

You will find attached the corrected manuscript.

Regards,

Nicolas Germain
